# Supernumerary B chromosomes of *Aegilops speltoides* undergo precise elimination in roots early in embryo development

Alevtina Ruban [1,2,6], Thomas Schmutzer [1,3,6], Dan D. Wu[1,4], Joerg Fuchs[1], Anastassia Boudichevskaia [1,2], Myroslava Rubtsova[1,5], Klaus Pistrick [1], Michael Melzer[1], Axel Himmelbach[1], Veit Schubert[1], Uwe Scholz [1] & Andreas Houben [1 ✉]

Not necessarily all cells of an organism contain the same genome. Some eukaryotes exhibit dramatic differences between cells of different organs, resulting from programmed elimination of chromosomes or their fragments. Here, we present a detailed analysis of programmed B chromosome elimination in plants. Using goatgrass *Aegilops speltoides* as a model, we demonstrate that the elimination of B chromosomes is a strictly controlled and highly efficient root-specific process. At the onset of embryo differentiation B chromosomes undergo elimination in proto-root cells. Independent of centromere activity, B chromosomes demonstrate nondisjunction of chromatids and lagging in anaphase, leading to micronucleation. Chromatin structure and DNA replication differ between micronuclei and primary nuclei and degradation of micronucleated DNA is the final step of B chromosome elimination. This process might allow root tissues to survive the detrimental expression, or over-expression of B chromosome-located root-specific genes with paralogs located on standard chromosomes.

[1] Leibniz Institute of Plant Genetics and Crop Plant Research (IPK) Gatersleben, 06466 Seeland OT Gatersleben, Germany. [2] KWS SAAT SE & Co. KGaA, 37574 Einbeck, Germany. [3] Martin Luther University Halle-Wittenberg, Institute for Agricultural and Nutritional Sciences, 06099 Halle (Saale), Germany. [4] Triticeae Research Institute, Sichuan Agricultural University, 611130 Wenjiang, China. [5] SAATEN-UNION BIOTEC GmbH, 06466 Seeland OT Gatersleben, Germany. [6] These authors contributed equally: Alevtina Ruban, Thomas Schmutzer. ✉email: houben@ipk-gatersleben.de

Throughout the life cycle of most organisms, the genetic information in all somatic cells stays unchanged. However, for a few exceptions the elimination of specific DNA is part of the developmental program. As such, programmed DNA elimination happens in diverse metazoa and unicellular ciliates and in the form of either one out of two principal ways: as a loss of chromosome fragments (observed in e.g., *Cyclops, Tetrahymena, Petromyzon* genera) or as a loss of entire chromosomes (in e.g., *Taeniopygia, Sciara, Acricotopus* genera) (reviewed by Wang and Davis[1]). In light of its wide phylogenetic distribution, programmed DNA elimination presumably evolved independently in different lineages. Divers hypotheses have been proposed to explain the significance of programmed DNA elimination including gene silencing, gene dosage compensation, mechanisms of sex determination, germline development and meiosis, and germline and soma differentiation[1,2].

To investigate programmed chromosome elimination in plants, *Aegilops speltoides* Tausch, a diploid grass with 7 pairs of chromosomes in its standard complement (A chromosomes), was analyzed. Compared with all recent wheat relatives, its genome is most closely related to the B-subgenome of wheat[3]. This species may carry up to 8 supernumerary B chromosomes (Bs) which are absent in the roots but stably present in the plant parts above ground in the same individual[4]. Bs are optional additions to the basic set of A chromosomes (As), and they occur in all eukaryotic groups[5]. They are assumed to represent a specific type of selfish genetic elements. The mechanism behind the tissue-specific distribution of *Ae. speltoides* Bs is unknown.

Here, we combine different approaches to gain insight into the enigmatic phenomenon of tissue-specific B chromosome distribution. We report that the B chromosome of *Ae. speltoides* contains gene-derived sequences, which are paralogous to genes on all 7 standard chromosomes and both cytoplasmic organellar genomes. The elimination of Bs is a strictly controlled and highly efficient root-specific process, which starts at the onset of embryonic tissue differentiation. Centromere activity independent micronucleation of Bs occurs due to chromosome nondisjunction during mitosis. Chromatin structure and replication differs between micronuclei and primary nuclei and degradation of micronucleated DNA is the final step in B chromosome elimination. We propose that some B-located gene sequences are expressable only in root tissues where their products are deleterious, or the elimination process is a product of selection for B chromosome maintenance in shoot tissue.

## Results

### Elimination of B chromosomes is strictly controlled.

B chromosome-carrying *Ae. speltoides* plants possess a constant number of B chromosomes in tillers, spikes, and leaves, while they are completely absent in roots[4,6]. To decipher the tissue-specific distribution of Bs, we first determined the location of Bs during embryogenesis. Flow cytometric analysis of nuclei isolated from individual, developing +B embryos revealed four distinct peaks representing 2C and 4C nuclei with and without Bs, indicating a chimeric genome composition (Fig. 1a). In contrast, only two peaks (2C and 4C nuclei without Bs) were present in 0B embryos. Thus, the tissue-specific elimination of Bs is initiated during early stages of plant development.

Next, to determine when and where B chromosomes are lost, tissue sections of embryos were hybridized in situ with the B-specific high-copy repeat AesTR-183. Undifferentiated embryos (3–5 days after pollination (DAP)) revealed B-specific signals in all cells (Fig. 1b, Supplementary Fig. 1a), suggesting that the Bs are mitotically stable during the first cell divisions after zygote formation. At the onset of embryo differentiation (6–8 DAP),

nuclei without B-specific signals, as well as B-positive micronuclei appear in the ventral side of the embryos, where later the meristematic zone will be formed (Supplementary Fig. 1b). At this stage, differentiation of pro-embryos into scutellum, coleoptile, shoot apical meristem and radicle primordium is initiated in cereals[7]. With further progression of embryo growth, the elimination of Bs from the root region proceeds and root cells of 15–17 DAP embryos are completely free of Bs (Fig. 1c). Micronuclei were observed in cells surrounding the root and these accumulated in the meristematic zone between the developing apical meristem and the embryonic root (Supplementary Fig. 1c). Despite the fact that B-positive micronuclei appeared also in other embryo parts, e.g. scutellum and coleorhiza, complete B chromosome elimination was observed only in the radicle and root cap (Fig. 1c). In seedlings, the transition zone between leaf and root is clearly marked by the loss of B-specific signals (Fig. 1d). Thus, the root-restricted elimination of Bs starts with radicle formation at the onset of embryo differentiation and occurs via micronucleation.

The loss of Bs also occurs in adventitious roots formed from tillering nodes (Supplementary Fig. 2a). However, the elimination process in this root-type is less efficient, as we observed scattered cell lineages in ~1.5 cm-long adventitious roots carrying B-specific signals (Supplementary Fig. 2b). Next, we demonstrated that the elimination of Bs also could be triggered by in vitro rhizogenesis (Supplementary Fig. 2c). Root formation was induced from undifferentiated calli, originating from the scutellum of immature +B embryos. Plants (*n* = 43) regenerated from +B calli revealed the same tissue-specific distribution of Bs as observed in seed-derived plants. Thus, the elimination of Bs is a strictly root-specific process, irrespective of the origin of the root.

To address whether the root-specific loss of Bs also occurs in different genetic backgrounds, pollen of +B *Ae. speltoides* plants was used to fertilize hexaploid wheat (*Triticum aestivum* Chinese Spring) with and without B chromosomes of rye[8]. In all hybrid combinations, the Bs of *Ae. speltoides* were stably present in leaves, but absent in roots. In contrast, the Bs of rye were always present in leaves and roots (Supplementary Fig. 3). Hence, the mechanism responsible for the elimination of *Ae. speltoides* Bs works in a hybrid background as well, but does not affect the coexisting Bs of rye.

### Nondisjunction results in the formation of micronuclei.

A well-known mechanism of chromosome elimination via micronucleus formation is based on the inactivation of centromeres[9]. To assay the centromere activity of Bs, we employed antibodies against the centromeric variant of histone H3 (CENH3), as a marker for active centromeres[10]. CENH3-positive, tubulin interacting, lagging B chromosomes were found in anaphase cells in the tissue where elimination of Bs occurs. The centromeres of lagging Bs were pulled towards opposite poles under the tension of microtubules (Fig. 2a–c, Supplementary Fig. 4a, Supplementary Movie 1). However, due to likely unresolved cohesion in the region of chromosome arms, Bs did not reach the poles at the same time as the A chromosomes. Lagging Bs were retained at the cell equator until the end of the division and finally entrapped in micronuclei (Fig. 2d, Supplementary Fig. 4b). Quantification of the CENH3 signal volumes showed no differences between As and Bs (Fig. 2e) suggesting that a decreased CENH3 amount is not responsible for the elimination of Bs. At telophase, when cell wall formation begins, the phragmoplast structures altered due to lagging Bs, which start to form micronuclei (Supplementary Fig. 4b). Interestingly, not all the Bs within a cell lagged at the same time; some segregated normally, while others stayed at the

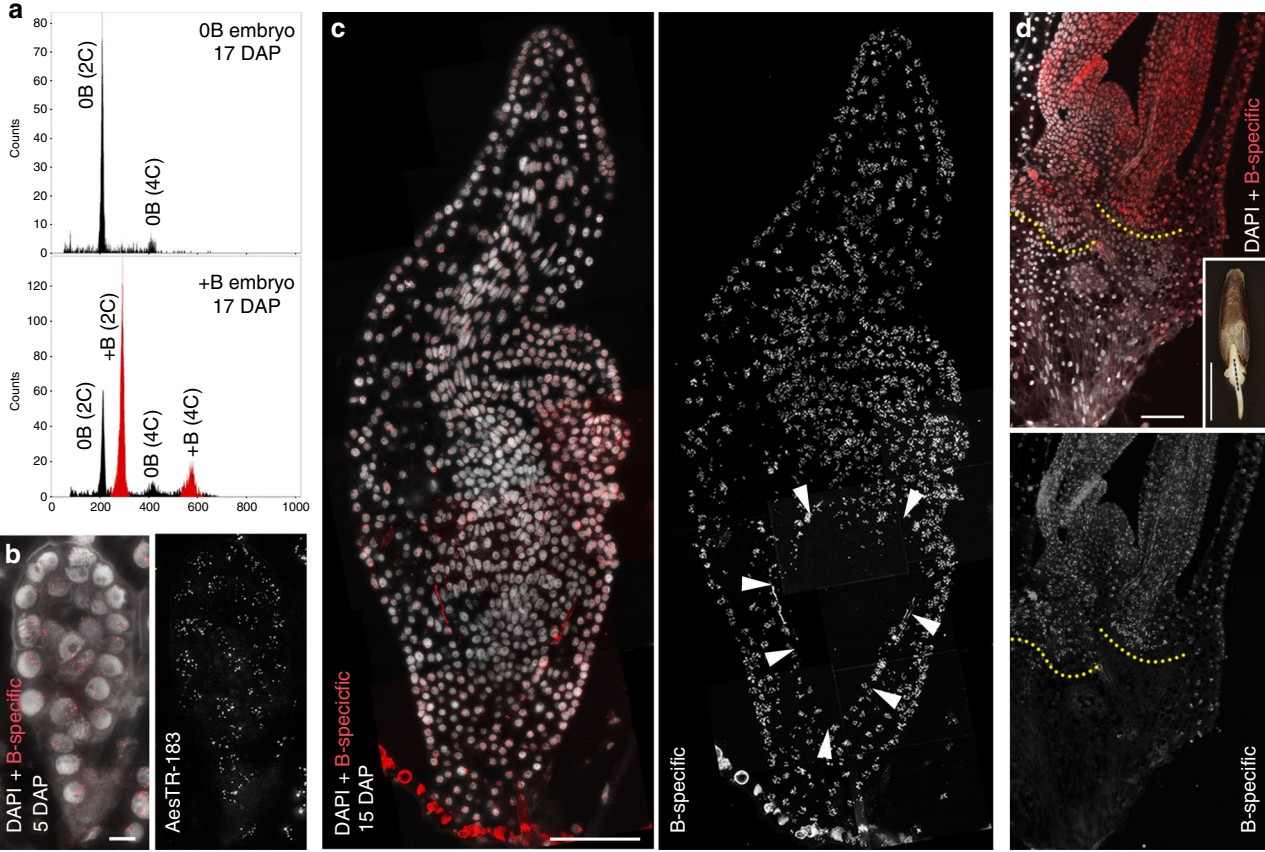

**Fig. 1 Root-restricted elimination of B chromosomes starts with radicle formation at the onset of embryo differentiation of _Ae. speltoides_. a** Flow cytometric histograms representing the DNA content of nuclei of 17 DAP 0B and +B embryos. Black peaks correspond to 0B nuclei with 2C or 4C DNA content and red peaks correspond to nuclei containing B chromosomes. The presence of additional 0B peaks in +B embryos implies that the elimination of Bs had been initiated prior to this stage of embryogenesis. Tissue sections (**b**, **c**) of +B embryos and (**d**) seedling after FISH using the B-specific probe AesTR-183. For each stage of embryo/plant development 2–4 embryos/seedlings were sectioned and analyzed. Obtained results were consistent.
**b** Embryo at the stage of 5 DAP exhibits B-specific signals in all cells. Scale bar, 10 µm. **c** 15 DAP old embryo exhibiting complete absence of Bs in root cells. Root region is indicated by arrowheads. Scale bar, 100 µm. **d** Tissue section of a one-day-old seedling. The transition zone between B-positive leaf and B-negative root tissues is marked by yellow dotted lines. Scale bar, 100 µm. The insert shows the seedling with the sectioning plane marked by a black dashed line. Scale bar, 5 mm.

cell equator. The presence of CENH3 signals in micronuclei is additional evidence that the centromeres of Bs are not inactivated during the elimination process.

**Degradation of micronuclei is the final step of elimination**. To analyse the final step in B chromosome elimination, the integrity of micronucleated DNA was tested by the terminal dUTP nick end-labeling (TUNEL) assay. 9% of micronuclei ($n = 184$) displayed TUNEL-signals (Fig. 3a), indicating that their DNA was strongly fragmented. No DNA cleavage was seen in the chromatin of primary nuclei. In line with the degradation process, none of the micronuclei ($n = 120$) incorporated EdU, despite normal DNA replication in the primary nuclei (Fig. 3b). Ultrastructural analysis showed that the double membrane in 20% of micronuclei ($n = 30$) was partially degraded (Fig. 3c, d), while the remaining micronuclei were surrounded by a double membrane as seen in primary nuclei (Fig. 3e, f). The higher chromatin staining intensity in 43% of micronuclei compared with the primary nucleus of the same cell indicates an increased degree of chromatin condensation (Fig. 3c, d). The high proportion of micronuclei with an intact double membrane and unfragmented DNA suggest a stepwise degradation process in micronuclei. Apparently, the fragmentation of micronucleated DNA is the final step of B chromosome elimination.

**Composition and evolution of the _Ae. speltoides_ B chromosome**. Programmed DNA elimination has been considered as a mechanism to silence genes in a tissue and developmental stage-specific way[1]. To test whether the B chromosome of _Ae. speltoides_ carries genic sequences and to identify its evolutionary origin, the sequence composition of the 570 Mbp large B chromosome was determined by comparative sequence analysis of total genomic DNA of 0B and +B plants and of microdissected Bs (Supplementary Table 1). A de novo assembly including both genomic 0B and +B data was performed based on ~438 million paired-end reads (genome coverage ~14.8-fold). The constructed genome sequence reached a total size of ~1.0 Gb, which is consistent with other draft whole-genome sequencing (WGS) assembly datasets using low-coverage paired-end reads[11–13]. The established assembly comprises 1,567,707 contigs with an N50 contig length of 1.3 kb. This allowed us to perform a prediction of gene models making use of the conserved coding sequences within the grass species _Brachypodium distachyon_, _Sorghum bicolor_, _Oryza sativa_, _Aegilops tauschii_ and the B subgenome of _Triticum aestivum_. In total, 82.1% of the 29,228 nuclear and organellar genes were assigned with a function, based on InterPro[14] and a BLASTP, using available gene information from the B subgenome of _T. aestivum_[15]. To identify the corresponding chromosomal locations of all A- and B-derived sequences we anchored 62.6% (981,908

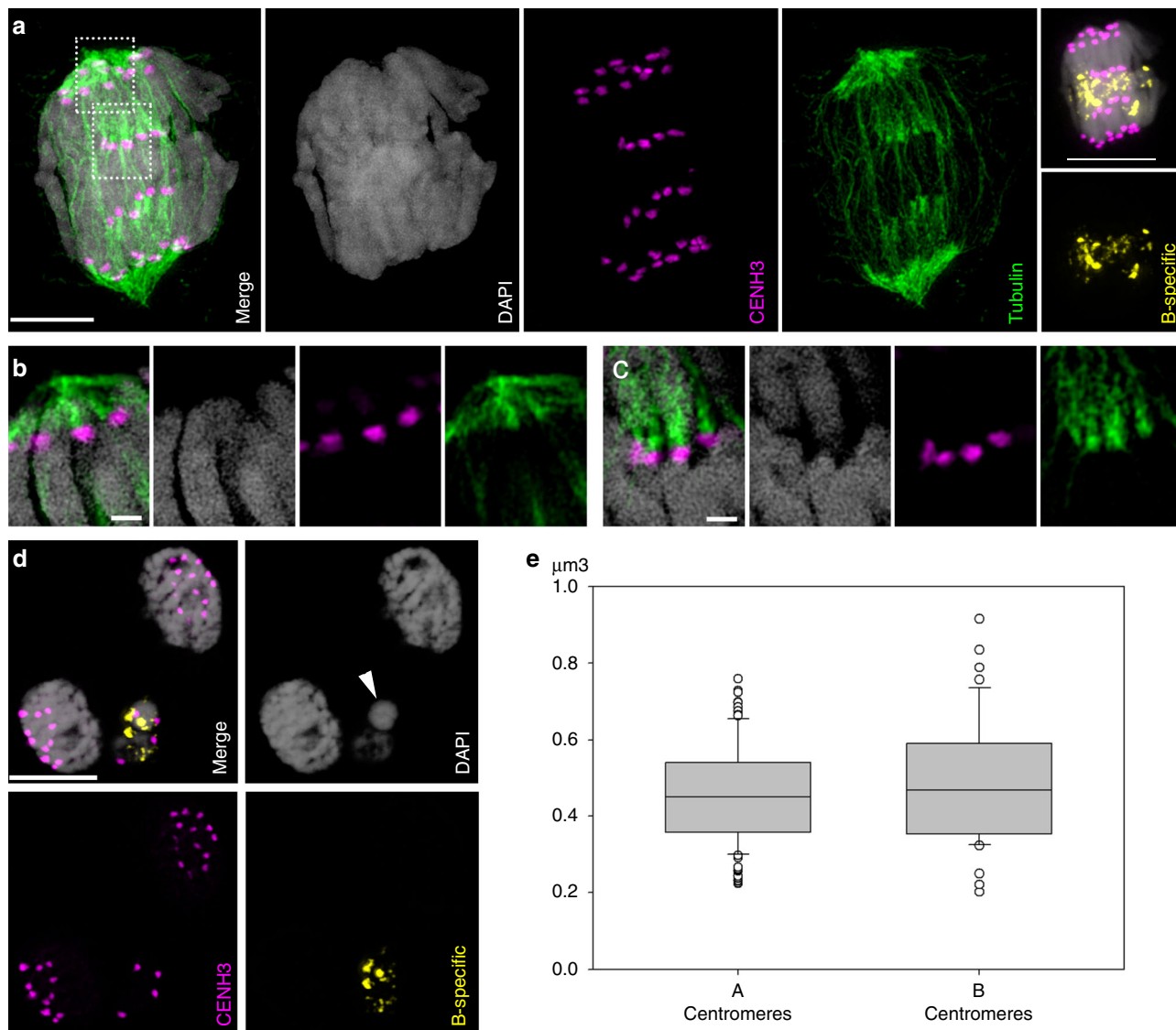

**Fig. 2 Nondisjunction of centromere-active B chromosomes results in the formation of micronuclei. a** Anaphase with lagging Bs after immunostaining of CENH3 (in purple) and α-tubulin (in green), for further details see Supplementary Movie 1. Scale bar, 5 μm. FISH signals of the B-specific probes AesTR-183 and AesTR-205 are shown in yellow. Scale bar, 10 μm. The enlarged regions (**b**, **c**) are marked by rectangles. Interaction of α-tubulin with the CENH3-positive centromeres of (**b**) A- and (**c**) B chromosomes. Scale bar, 1 μm. In total 7 anaphase cells with lagging B chromosomes were analyzed in 3 embryos using super resolution microscopy. On 5 from those cells interaction of α-tubulin with the CENH3-positive centromeres was consistently observed. **d** Formation of B-containing micronuclei during telophase. Note the position of centromeres on the opposite sides of the micronuclei resulting from the spindle microtubule tension during anaphase. The centromeric signals were observed for 60 micronuclei in 10 embryos. Scale bar, 10 μm. **e** Box plot showing CENH3 signal volumes (Y-axis, μm³) on A- and B centromeres at anaphase. Measurements are performed for 7 anaphase cells and data are statistically treated. Because the data failed the normality test, a Mann–Whitney test for comparison of both centromere types was performed. The data were not significantly different ($P = 0.363$). The box indicates the 25th–75th percentiles, the line marks the median. The error bars indicate the 10th and 90th percentiles and the open circles the outlying points. Source data underlying **e** are provided as a Source data file.

contigs) of the assembled contigs to the pseudomolecules of the *T. aestivum* B subgenome.

To reconstruct the B chromosome origin we analyzed the WGS assembly using two independent validation methods based on (1) the sequence reads of microdissected Bs (MiDiSeq) and (2) the coverage ratio analysis approach[16], using *k*-mer frequency ratios calculated by *Kmasker plants*[17]. For the first approach, we used quality enriched reads from microdissected Bs aligned with the constructed WGS assembly to confirm sequences from Bs. For the second approach, we filtered B-derived contigs, which were characterized by significantly increased *k*-mer values in the +B dataset in relation to the *k*-mer values obtained in the 0B dataset.

In total, 174,490 contigs (~90 Mbp, representing 15.8% of the B chromosome) of low, conservative, moderate and high confidence, representing confidence classes 1–4, respectively, were assigned to the B of *Ae. speltoides* (Supplementary Table 2). The highest confidence class was assigned to 948 contigs (696 kbp), which were validated by MiDiSeq and showed significant 5-fold change differences in the *k*-mer frequency ratio analysis. We observed an increase in the contig number (moderate: 7,713; conservative: 22,369 and weak: 143,460) inverse to the decrease of confidence (Supplementary Table 3). This explains that the reliability of our assignments to the B chromosome is very strong for a subset of 31,030 contigs (class 1–3) and if less stringent

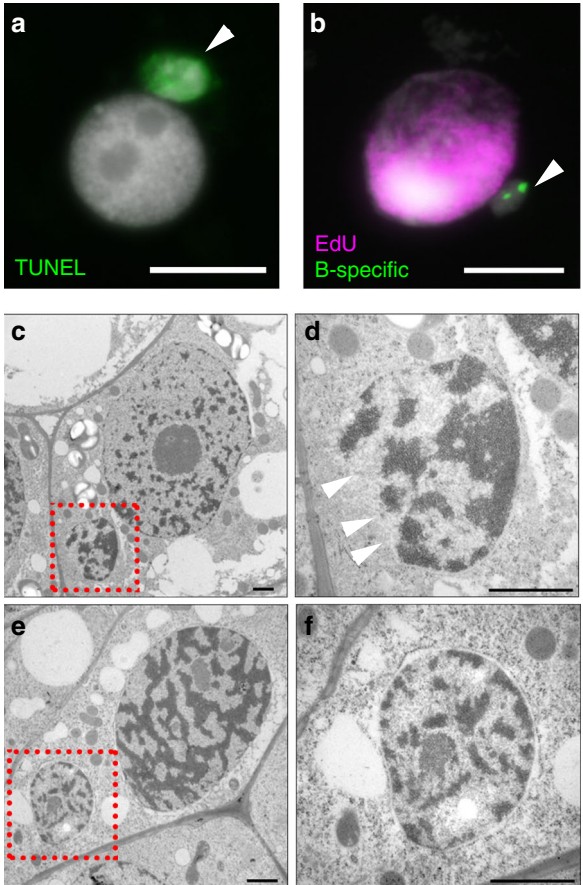

**Fig. 3 Chromatin structure and replication differs between micronuclei and primary nuclei. a** Micronuclei chromatin assessed by the TUNEL assay. 9% (*n* = 184) of analyzed micronuclei (arrowed) were TUNEL positive (in green). Scale bar, 10 μm. **b** None of the micronuclei (*n* = 120) incorporated EdU, despite normal DNA replication of the primary nuclei. EdU is shown in purple, the B-specific FISH probe AesTR-183 in green. Scale bar, 10 μm. **c**–**f** Electron micrographs of ultra-thin tissue sections of developing +B embryos. Micronuclei (*n* = 30) are marked by red squares and are further enlarged. **c**, **d** A micronucleus with a disrupted membrane (arrowed, was observed in 20% of the cases) and an increased chromatin density (43% of micronuclei showed dense chromatin) in comparison to the corresponding nucleus. Scale bar, 1 μm. **e**, **f** Micronucleus surrounded by an intact double membrane and with a considerable amount of euchromatin. Scale bar, 1 μm.

thresholds are used the assignment quantity can be further increased.

35.2% of the B chromosome-derived contigs were assigned to the B subgenome of wheat. Our results show that all A chromosomes were involved in the origin of the B chromosome (Fig. 4). Chromosome 5 has the highest contribution (7.2%) and chromosome 4 (2.5%) the lowest contribution (Supplementary Table 4). In line with previous FISH experiments[6], the B of *Ae. speltoides* showed strong enrichment with organelle-derived DNA. 3.0% of the ~2 million uniquely aligned reads from the microdissected B originated from organelle DNA. Almost half of the mitochondrial genome (61 contigs representing 219.5 kbp) and a 35.3 kbp long contig with similarity to the chloroplast genome of *Ae. speltoides* were found in the set of B chromosome-assigned sequences.

The repeat fraction was studied in (1) the total WGS assembly and (2) B chromosome-assigned contigs using RepeatMasker[18] utilizing a wheat repeat library. The LTR elements from *Gypsy* and *Copia* families account for the highest proportion of repeats

in the total assembly (13.44%) and these also were found with high abundancy among the B chromosome-assigned contigs (4.56%). However, while the *Gypsy* elements are twice more abundant than *Copia* elements in the total assembly, in the B assigned as well as in B-specific contigs, *Copia* elements became predominant (Supplementary Tables 5, 6).

Copy-number analysis was performed on a subset of 11,785 repetitive elements (length > 30 bp, minimal avg. *k*-mer count 10 in 0B dataset) derived from the 35,583 repeats predicted by RepeatMasker in the B chromosome-assigned contigs. We compared *k*-mer frequencies between the 0B and +B datasets and observed a clear tendency of *k*-mer frequency increase (~20.8%) for B-derived sequences in the +B dataset. 43 repeat elements doubled their abundancy via B-specific amplification. The overall abundance of all B assigned contigs (11,937) that were classified as mathematical repeats (avg. *k*-mer count >50) increased by 18.9% in +B dataset in comparison to the 0B dataset and 384 contigs had more than doubled their occurrence. At the gene level, we found that the average *k*-mer ratio between the 0B and +B datasets for B-assigned genes was increased to almost 2-fold, suggesting an amplification of B-located genic sequences (Supplementary Fig. 5).

In addition to sequences shared by A and B chromosomes, we identified 2792 contigs (total length 3.84 Mbp, 4.3% of B-derived sequences) that were characterized in silico as B chromosome-specific. 90.3% of these (2521 contigs with a total length of 3.67 Mbp) were found in the confidence class 3, and 9.7% (271 contigs with a total length of 164 kbp) were found in the highest confidence class 4. The B-specificity of a sequence subset was confirmed by genomic PCR and FISH (Supplementary Fig. 6).

**The B chromosome of *Ae. speltoides* contains genic sequences.**
229 genes with an average length of 1402 bp (CDS) were identified in B chromosome-assigned contigs of the confidence classes 2–4. Most (75%) of these B chromosome genic sequences were assigned to the seven A chromosomes of the wheat B subgenome (Fig. 4; Supplementary Data 1). In line with the proportion of contigs assigned to the B chromosome, wheat chromosome 3 (35 contigs) contributed most and chromosome 4 (17 contigs) contributed the smallest number of genes (Supplementary Table 4). This set of in silico-validated B genic sequences contains complete genes, including start and stop codons, of mitochondrial and chloroplast origin (13 and 8 respectively). Sequence similarity analysis was performed using B-located paralogs that were identified by alignment of microdissected sequences to the 0B assembly. B-located paralogs were aligned against the merged (0B and +B) WGS dataset. The sequences that had a paralogous counterpart in the merged (0B and +B) WGS dataset were of 4.94 Mbp with minimal 5-fold sequence coverage. In these B-located paralogs, we performed SNP calling using vcftools[19] and sequence data from +B and 0B. We observed higher SNP counts in B-specific sequences of the microdissected reads. This increase of SNP density (31 bp in +B versus 38 bp in 0B) can be explained because the redundant B chromosomal copy can accommodate a higher mutation load than its A related pendant. Furthermore, we found that B chromosome genic sequences are more conserved and have a lower SNP density (42 bp) (Supplementary Table 7).

The GO enrichment analysis of 229 genes assigned to the B chromosome of *Ae. speltoides* was based on the retrieved GO terms from two annotation approaches: InterPro (45 GO term annotated genes) and the TGACv1 wheat genome annotation (95 GO term annotated genes). In the biological process category, genes involved in nucleobase, nucleoside, nucleotide and nucleic acid metabolic process (GO:0006139) and nitrogen compound metabolic process (GO:0006807) were overrepresented according

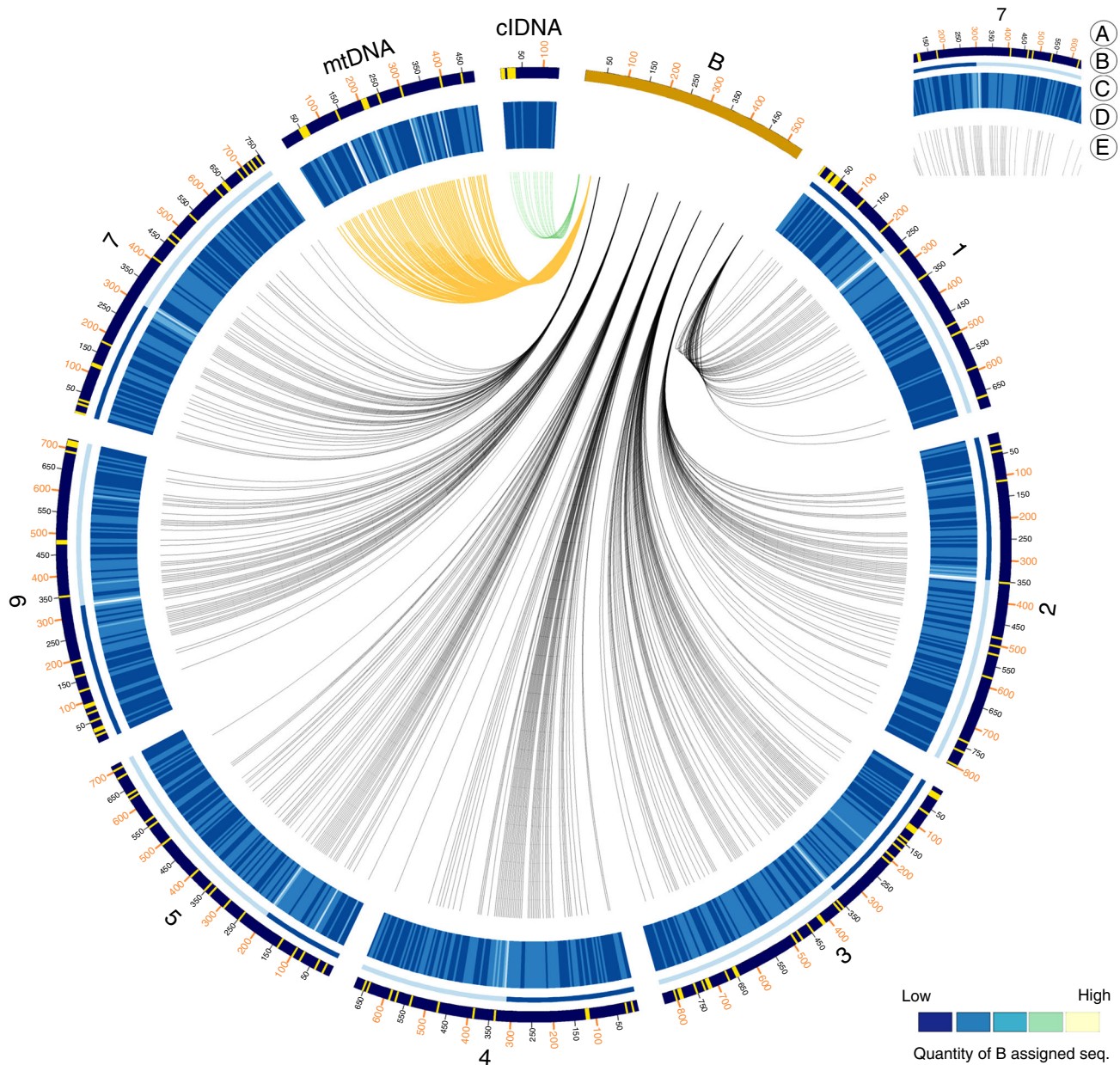

**Fig. 4 Multichromosomal origin of *Ae. speltoides* B chromosomes.** Visualisation generated with Circos[61]. For clarity and visual aspects, the size of both, mitochondrial and chloroplast organelles, are depicted with a ×1000 magnification. The heatmap illustrates B assigned sequence segments, derived from the number of aligned microdissected reads. All other chromosomes are depicted at real Mb length and are analyzed in 5 Mb window segments. The order of B-assigned sequence segments were not resolved: the respective segments are constituted as aggregated blocks. (A) Chromosome IDs and Mb scales (B) Gene locations (yellow) in 5 Mb segments (C) Centromere locations (D) Heatmap (quantity of B assigned seq. bp) in 5 Mb segments (E) Linked segments of B origins. Source data are provided as a Source data file.

to both approaches. Functional annotation based on the InterPro approach allowed also detection of GO terms DNA metabolic process (GO:0006259), DNA integration (GO:0015074), and chromosome organization (GO:0051276). In addition, GO terms related to cellular biosynthetic process (GO:0044249), biosynthetic process (GO:0009058), and generation of precursor metabolites and energy (GO:0006091) were significantly over-represented (Supplementary Fig. 7a) according to the data based on the TGACv1 wheat genome annotation.

We detected also increased incidence of GO terms in the category molecular function, most of which originated from the TGACv1 wheat genome annotation. The GO terms related to binding, such as ion binding (GO:0043167), zinc ion binding

(GO:0008270), transition metal ion binding (GO:0046914), and metal ion binding (GO:0046872) were enriched. Another category of overrepresented GO terms relates to peptidase activity. These are such terms as aspartic-type peptidase activity (GO:0070001), peptidase activity (GO:0008233), endopeptidase activity (GO:0004175). The GO term nucleic acid binding (GO:0003676) was enriched among 229 genes according to both approaches (Supplementary Fig. 7b).

In view of the strict elimination of B chromosomes in roots it was pertinent to ask whether any of the B-located genes were specifically expressed in roots. In order to ask whether some of the B chromosome-located genic sequences represent potential root-specific genes we aligned the 229 B-assigned specific genes

with all available transcript sequences of high-confidence genes during eight developmental stages of barley[20,21] (Supplementary Fig. 8, Supplementary Data 2 and 3). Notably, 20 B-assigned genic sequences (~10%) revealed root-specific expression patterns. Root-specific expression was confirmed by qRT-PCR for 4 paralogous A chromosome-encoded genes (Supplementary Fig. 9). Hence, the root-specific elimination of Bs results in the ultimate silencing of genes potentially involved in root function. These genes could have the potential to influence root development in *Ae. speltoides* +B plants, particularly if they are present at higher than normal copy numbers as they would be if B chromosomes were present in the root.

## Discussion

Our study demonstrates that programmed elimination of chromosomes exists in plants. We provide direct insights into when and how the tissue-specific process of B chromosome elimination occurs in the goatgrass *Ae. speltoides*. With the onset of embryonic tissue differentiation, which occurs ~7 days after pollination in grasses[7], Bs undergo elimination in cells destined to become root-cells. The organ-type specificity of this process is supported by the absence of Bs in adventitious roots and roots developed from B-positive undifferentiated calli. The exclusive elimination of *Ae. speltoides* Bs in roots of hybrids derived from crosses between wheat possessing additional rye Bs and +B *Ae. speltoides* implies that the elimination process is B chromosome-type specific. Hence, the observed chromosome elimination process is not a result of genome instability but is a specific, strictly-controlled process.

There is a striking similarity between programmed elimination of supernumerary germ-line-restricted chromosomes in some insect species[22,23] and the Bs of *Ae. speltoides*. In both scenarios, supernumerary chromosomes fail to separate their chromatids and linger at the cell equator during mitotic divisions at the stage of embryo development. One may speculate that the underlying mechanism is widespread and evolutionarily conserved across all the eukaryotes, from animals to plants. Based on the observed interaction between the CENH3-positive centromeres and tubulin of both chromosome types, we conclude that the centromeres of lagging Bs are actively involved in mitotic division. However, at anaphase, in contrast to the separated A chromatids, the chromatids of Bs (except their pericentromeric regions) remain cohesive, likely due to a delayed release of sister chromatid cohesion. Notably, despite this unusual lagging of Bs, the cell cycle progresses essentially normally. Whether the mitotic spindle assembly checkpoint is impaired or Bs escape the checkpoint control remains unknown.

Similarly to the fate of micronuclei containing paternal chromosomes of unstable intraspecific hybrids[24], the elimination of +B micronuclei is a step-wise process. None of the DNA in micronuclei undergoes replication although the corresponding nucleus of the same cell proceeds through S-phase. According to our data, the micronuclei are not degraded immediately after their formation as the number of micronuclei with intact chromatin and a double membrane was higher than those showing signs of DNA fragmentation. A reverse process of micronuclei inclusion back into the primary nucleus is unlikely, as a complete absence of Bs in roots was observed at the end of embryogenesis.

Comparing the cellular process of post-meiotic B chromosome drive in *Ae. speltoides*[25] and rye[26] with the process of B elimination described herein reveals striking similarities. B chromosomes are often preferentially inherited, deviating from Mendelian segregation. The balance between the so-called chromosome drive and the negative effects that the presence of Bs applies to the fitness of their host determines the frequency of Bs

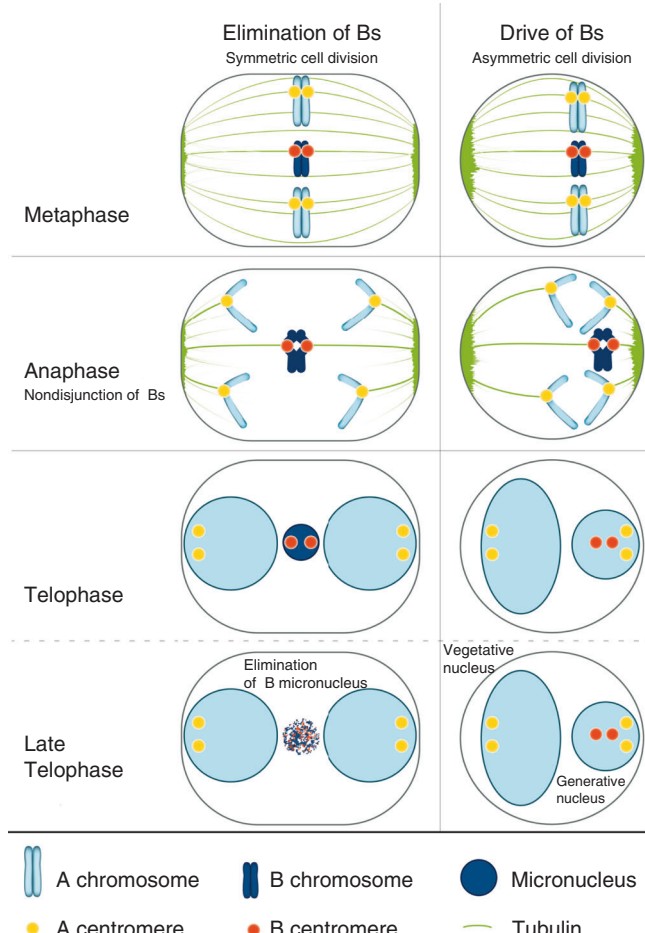

**Fig. 5 Schematic comparison of the cellular process of B elimination with the process of post-meiotic B chromosome drive reveals striking similarities.** In both processes, nondisjunction of Bs occurs despite centromere activity and centromere-tubulin interaction. Spindle symmetry differs between the two processes: during the first pollen mitosis an asymmetric cell division occurs whereas the spindle in roots is symmetrical. Therefore, in roots, lagging B chromosomes form micronuclei and undergo elimination. In contrast, due to the asymmetric geometry of the spindle at first pollen mitosis, the inclusion of the lagging joint B chromatids in the generative nucleus takes place and chromosome accumulation occurs.

in a particular population[27]. As summarized in Fig. 5, in both processes nondisjunction of Bs occurs despite centromere activity and centromere-tubulin interaction. Spindle symmetry differs between the two processes: during the first pollen mitosis an asymmetric cell division occurs whereas the spindle in roots is symmetrical[25]. As a consequence, in roots, lagging B chromosomes form micronuclei and undergo elimination. In contrast, due to the asymmetric geometry of the spindle at first pollen mitosis, the inclusion of the lagging joint B chromatids into the generative nucleus takes place and chromosome accumulation occurs. It is probable that the type of spindle organization (symmetric versus asymmetric) determines whether drive or elimination of B chromosomes becomes the consequence.

It must be remembered that elimination and drive of Bs are specific mechanisms acting exclusively in particular types of tissue, namely in proto-root cells and in pollen grains at the first mitotic division. The strict absence of Bs in the roots of *Ae. speltoides* and other species suggests that care should be taken regarding the determination of chromosome number based on one tissue-type only, particularly as roots are often used to

determine the chromosome number of a plant species. Therefore, it would be best to determine chromosome numbers and genome sizes based on different organs including generative tissue of multiple individuals of different accessions if working with a less-studied species.

Future understanding of the molecular mechanism behind tissue and chromosome type-specific cohesion dysfunction may provide clues about the process of chromosome nondisjunction, which is a major cause of genetic diseases across species. Clearly, chromosome-specific elimination is a highly efficient process and its understanding could help to unravel the mechanism by which undesirable DNA is eliminated from the host genome. This also could lead to novel biotechnological applications and, importantly, begin to understand how errors occur during cell division in human and other organisms.

Programmed chromosome elimination is clearly a mechanism for the ultimate silencing of genes, with expression that may be undesirable and potentially harmful in some somatic tissues[2,28]. However, in comparison to animals in which germline cells are differentiated during early embryogenesis and persist throughout the entire life, in plants there is no defined germ-cell lineage present until the switch to generative organ formation from undifferentiated stem cells[29]. Based on our sequence analysis and previous reports about the origin of Bs, B chromosomes are by-products of host genome evolution, which implies the presence of A chromosome-derived and organelle genome-derived gene duplicates on Bs[30–32]. Assuming that those gene duplicates may either retain functionality or develop paralogous functions, elimination of Bs in roots may represent either dosage compensation or an extreme and irreversible mechanism for silencing B-located genes, contributing to the distinction between root and shoot. In addition, one may assume that B chromosome elimination serves to spare root cells the costs of replicating and maintaining large quantities of unnecessary DNA. *Ae. speltoides* carries up to eight B chromosomes, each representing around 10% of the standard genome. Our current analysis of the origin and composition of the *Ae. speltoides* B chromosome revealed the existence of 229 B-located genic sequences. However, our analyses do not allow any conclusion regarding the integrity and functionality of these B-located genes. We suggest that root-specific elimination of Bs is required to remove B chromosome-located genes which, if expressed in roots, would lower the inheritance of the selfish chromosome because of reduced plant vigor. The possibility that positive selection for B chromosome maintenance in shoot tissues has occurred remains to be investigated. A multi-chromosomal origin of the *Ae. speltoides* B chromosome is supported by the many sequences that are similar to different regions of the A chromosomes. A comparable amalgamation of diverse A-derived sequences has been previously postulated for the Bs of e.g. rye[30], cichlid fish[16], red fox and raccoon dog[31] and grasshoppers[33]. We also observed that large amounts of mitochondrion- and chloroplast-derived DNA are integrated into the B of *Ae. speltoides*. It seems likely that repeated integration of organelle-derived DNA occurred due to the reduced selection pressure on B chromosomes.

## Methods

**Plant material and crosses**. *Ae. speltoides* Tausch from Tartus, Syria (PI 487238; USDA-ARS, Aberdeen, Idaho, USA) and from Katzir, Israel (TS 89: Institute of Evolution collection, Haifa, Israel) plants with and without B chromosomes were utilized. Plants were grown at the IPK Gatersleben (herbarium accessions GAT 52666-52674 and GAT 52656-52665; 52675-52676, respectively) under greenhouse conditions: 16 h light, day temperature 20–24 °C, night temperature 17–19 °C. At the stage of 3 leaves, plants were kept at 12–15 °C for approx. 1 month to ensure better tillering and synchronous flowering. Cross-pollination between both populations was prevented. To obtain embryos at a precisely defined developmental stage, spikes were emasculated a few days before flowering and manually pollinated

once with fresh pollen. For an approximate staging of embryos, the day of anthesis was marked and spikes were left for free pollination. In crossing experiments with *Ae. speltoides* +Bs, either *Triticum aestivum* L. Chinese Spring (TRI 12922 from IPK-Gatersleben Genebank) or Chinese Spring with added rye Bs[8] were used as female plants. When tillering started, wheat plants were kept at 4 °C for approx. 1 month to synchronize their flowering time with that of the pollen donor plants. Wheat spikes were manually emasculated a few days before flowering and pollinated with fresh *Ae. speltoides* pollen. Growth conditions for wheat and the resulting hybrids were the same as described for *Ae. speltoides*.

**Organogenic calli production and plant regeneration**. Tissue culture was carried out based on previous reports[34,35]. Briefly, *Ae. speltoides* spikes containing 11–17-day old kernels were rinsed with distilled water, placed in 70% ethanol for 1 min, rinsed again with distilled water and surface sterilized in 1% Na-hypochlorite for 10 min. After washing the spikes with distilled water three times for 5 min, the embryos were isolated from kernels under sterile conditions and placed on callus induction medium with the scutellum directed up in Petri dishes. The medium contained mineral salts of MS with vitamins, 2 mg/l 2,4-dichlorophenoxyacetic acid, 30 g/l sucrose, 0.5 g/l casein hydrolysate, 5 μM CuSO$_4$ and 6.5 g/l Gelrite. After three weeks in darkness at 24 °C, calli were transferred to regeneration medium and cultivated under long-day conditions (16 h light (3000 Lux)/ 8 h darkness) at 24 °C. The medium was composed of mineral salts of Q & L medium including vitamins, 1 mg/l kinetin, 3.5 mg/l zeatin riboside, 20 g/l maltose, 1 g/l casein hydrolysate, 5 μM CuSO$_4$, and 3.3 g/l Gelrite. One-month-old regenerants were transferred to MS medium with vitamins, 1 mg/l thiamine hydrochloride, 0.1 g/l myo-inositol, 30 g/l sucrose, 0.1 g/l casein hydrolysate, 5 μM CuSO$_4$ and 3.3 g/l Gelrite. After 5 weeks on the medium under long-day conditions (16 h light (3000 Lux)/ 8 h darkness) at 24 °C, the plants were transferred to soil in a greenhouse. All the plant growth hormones and media components used in this study were purchased from Duchefa Biochemie, The Netherlands.

**Isolation of genomic DNA and microdissection of chromosomes**. Genomic DNA was isolated from leaf and root tissue with the DNeasy Plant Mini Kit (Qiagen, Germany). In order to isolate B chromosome-derived DNA, micro-dissection of Bs was performed after chromogenic in situ hybridization (CISH) with the B-specific repeat AesTR-183[25]. This probe was labeled with digoxigenin-11-dUTP (Roche, Germany) using the Nick Translation Mix (Roche, Germany). The application of CISH allowed the usage of a microdissection system equipped only with a bright-field microscope. Meiotic metaphase I chromosomes of a +5B *Ae. speltoides* plant were prepared by squashing the meiocytes between two coverslips (24 × 50 mm and 22 × 22 mm) in a drop of 45% acetic acid after 30 min fixation of anthers in 3:1 (ethanol: acetic acid). The smaller coverslip was removed after freezing the preparation in liquid nitrogen and the specimens were kept at room temperature in 96% ethanol. Prior CISH, slides were air-dried at room temperature. Labeling of B chromosomes was performed using the ZytoDot CISH Implementation Kit (ZytoVision GmbH, Germany). However, steps involving xylene and pepsin treatment were omitted and the procedure started after heat pretreatment which was done in a microwave for 1 min at 800 W.

Microdissection of chromosomes was performed using an inverted microscope Zeiss Axiovert 35 (Carl Zeiss GmbH, Germany) equipped with a micromanipulator (5170, Eppendorf, Germany). Selected chromosomes were isolated from the cover slip with a glass needle and transferred into the tube with 1 μl droplet containing proteinase K (Boehringer, 0.5 mg/ml) in 10 mM TRIS-HCl, pH 8.0, 10 mM NaCl and 0.1% (w/v) SDS.

**Preparation of tissue sections**. Tissue sections of developing *Ae. speltoides* kernels, 1-day old seedlings, adventitious roots, and organogenic calli were prepared according to the previously published protocol[36]. Briefly, specimens were fixed in freshly-prepared ice-cold 3% paraformaldehyde for 5–7 h at 4 °C and infiltrated with a series of polyester wax/ethanol solutions with increasing wax concentration (1/2, 1/1, 2/1 v/v) and in pure wax for 12–24 h in each solution. After infiltration material was embedded in pure wax using silicone casting molds (Plano, Germany). Polyester wax was composed of 9 parts of poly(ethylene glycol) distearate ($M_n$ = 930) and 1 part of 1-hexadecanol (w/w). 10 μm thick tissue sections were cut with a Leica RM2265 microtome (Leica Biosystems, Germany) equipped with low profile Leica 819 microtome blades (Leica Biosystems, Germany) and spread on the slide with 1 μl drop of distilled water. Sections were dried overnight at room temperature, dewaxing was performed by washing the slides 2 × 10 min in 96% and 2 × 10 min in 90% ethanol. Dewaxed slides were immediately transferred into 1× PBS solution and used for immunostaining and FISH.

**Preparation of mitotic chromosomes and squashed embryos**. Apical meristems of 3 days old seedlings were fixed in 3:1 (ethanol: acetic acid) for one day at room temperature. After washing in water and 0.01 M citrate buffer, meristems were digested with 1.4% cellulase, 2% pectolyase, and 1% cytohelicase in 0.01 M citrate buffer for 1 h at 37 °C. Cell suspension was prepared and dropped onto slides according to Aliyeva-Schnorr et al.[37]. Briefly, after enzymatic treatment meristems were washed first in 0.01 M citrate buffer, then in 96% ethanol. Ethanol was replaced with 75% acetic acid: 25% ethanol solution and meristems were

disintegrated using a plastic pistil. The obtained suspension was dropped onto the cold slides placed on a hot plate under high humidity condition. After drying, preparations were stored in 96% ethanol at −20 °C until use for fluorescence in situ hybridization (FISH). For sequential immunostaining and FISH, squashes of 10–15 days after pollination (DAP) embryos were prepared as following. Isolated embryos were fixed in freshly-prepared 3% paraformaldehyde in 1× PBS on ice for 1 h. The embryos were then squashed under a coverslip in a drop of 1× PBS without preceding enzymatic treatment. Coverslips were removed after freezing the slides in liquid nitrogen and the preparations were immediately transferred to 1× PBS.

**Immunostaining and FISH**. Immunostaining of dewaxed tissue sections and squashed embryos was done according to Badaeva et al.[36] and Houben et al.[38]. Slides were microwaved for 1 min at 800 W in 10 mM citrate buffer to improve the accessibility of the chromatin. After blocking with 5% bovine serum albumin in 1× PBS, slides were incubated with primary antibodies for 24–48 h at 4 °C and with secondary antibodies for 1 h at 37 °C. Mouse anti-α tubulin (Sigma-Aldrich, USA, cat. no. T 9026, dilution 1:200) and rabbit anti-grass CENH3 Sanei et al.[39] (dilution 1:3000) were the primary antibodies and anti-mouse Alexa 488 (Molecular Probes, USA, cat. no. A11001, dilution 1:200) and anti-rabbit rhodamine (Jackson ImmunoResearch, USA, cat. no. 111-295-144, dilution 1:600) were used as secondary antibodies.

For FISH of immunolabeled tissue sections: after immunostaining, the slides were fixed in 3:1 (ethanol: acetic acid) for 10 min and washed 2 × 5 min in 2×SSC. Plasmid DNA of clones AesTR-183, AesTR-205, AesTR-148[25] and pTa-794[40] was isolated with the QIAGEN Plasmid Midi Kit and labeled with dUTP-ATTO-488, dUTP-ATTO-550 or dUTP-ATTO-647N by nick-translation using the ATTO NT Labeling Kit (Jena Bioscience, Germany). FISH was performed according to Badaeva et al.[36]. Slides were treated with 4% formaldehyde solution for 10 min and washed 3 × 5 min in 2×SSC, then dehydrated in ethanol series 70, 85, 90% for 1 min in each. Probes were denatured at 95 °C for 10 min in hybridization mixture (1 g of dextran sulfate dissolved in 1 ml ddH2O, 5 ml deionized formamide, 1 ml 20×SSC, 1 ml fish sperm) and cooled down on ice until use. For 20 µl of hybridization solution 1 µl of each probe was used and the rest was filled up by hybridization mixture. Hybridization solution was applied on dry slides and covered with coverslip. Slides were denatured at 80 °C for 1 min (5 min for tissue sections) on a hot plate. Hybridization was performed in a moist chamber at 37 °C for 24 h. After post wash in 2×SSC at 55–60 °C for 20 min, slides were dehydrated in ethanol series 70, 85, 90% for 1 min in each, air-dried and mount with Vectashield (Vector Laboratories, USA) + 4′,6-diamidino-2-phenylindole (DAPI) mixture.

**TUNEL assay**. To test the integrity of micronucleated DNA, 10–15 DAP embryos were probed with the Click-iT™ Plus TUNEL Assay (Invitrogen, Germany) according to manufacturer's instructions. Embryos were prepared as described for immunostaining. As a positive control, specimens were treated with 1 U of DNase for 30 min at room temperature. To confirm the staining specificity, a negative control was performed by excluding the TdT enzyme.

**Analysis of DNA replication**. EdU (5-ethynyl-2′-deoxyuridine) incorporation was performed according to Klemme et al.[41] using the EdU Cell Proliferation Kit (Baseclick, Germany). 10–15 DAP embryos were incubated for 3 h with a 15 µM EdU solution, followed by water for 3 h and fixation in 3:1 (ethanol: acetic acid) for one day at room temperature followed by several days at 4 °C. After squashing the embryos in 45% acetic acid, the click reaction was performed to detect EdU according to the kit protocol. For subsequent FISH, slides were washed in 2× SSC for 5 min, postfixed in 3:1 (ethanol: acetic acid) and the further steps were carried out as described above in immunostaining and FISH section.

**Light and super-resolution microscopy**. Classical fluorescence imaging was performed using an Olympus BX61 microscope equipped with an ORCA-ER CCD camera (Hamamatsu, Japan). Images were acquired in gray scale with the software CellSens Dimension 1.11 (Olympus Soft Imaging Solutions, Germany). Images of tissue sections were taken by Z-stacks for each fluorescence channel separately. For embryos older than 7 DAP, several overlapping pictures were necessary to acquire entire embryos. Deconvolution of Z-stacks was performed with the Nearest Neighbor algorithm. Final images were pseudocolored and merged in Adobe Photoshop CS5 (Adobe Inc., USA). The levels and curves tools were used to adjust brightness and contrast of the images. To analyse the ultrastructure and spatial arrangement of signals and chromatin at a lateral resolution of ~120 nm (super-resolution, achieved with a 488 nm laser), 3D structured illumination microscopy (3D-SIM) was applied using an Elyra PS.1 microscope system equipped with a Plan-Apochromat 63×/1.4 oil objective lens and the software ZENblack (Carl Zeiss GmbH, Germany). Image stacks were captured separately for each fluorochrome using 561, 488, and 405 nm laser lines for excitation with appropriate emission filters[42]. Maximum intensity projections of whole cells were calculated via the ZEN software. The image stacks were used to measure the CENH3 signal volumes of A and B chromosomes after surface rendering via the Imaris 8.0 (Bitplane, Switzerland) software. A 3D-SIM stack was also used to produce movie 1 by Imaris 8.0.

The data of CENH3 signal volume measurements were statistically treated. Because the data failed the normality test, a Mann–Whitney test for comparison of both centromere types was performed. The data were not significantly different (P = 0.363). The box in the box plot (Fig. 2c) indicates the 25th–75th percentiles, the line marks the median. The error bars indicates the 10th and 90th percentiles and the open circles the outlying points.

**Transmission electron microscopy**. High pressure freezing (HPF) with a Leica EMHPF (Leica Microsystems, Germany) for cryofixation of 15–17 DAP embryos was used to minimize structural alterations during sample preparation. Isolated embryos were loaded into aluminium platelets with a cavity of 0.2 mm and a paste of 1:1 (v/v) mixture of yeast (Arxula adeninivorans) and cyanobacteria (Synechocystis 6308) as filler. After covering with the flat surface of a 0.30 mm platelet, samples were frozen under a pressure of ~2000 bar and transferred into an automated freeze substitution (FS) unit (Leica Microsystems, Bensheim, Germany). FS and resin embedding were carried out as described in Supplementary Information (Supplementary Table 8). Ultrathin sections (70 nm) were cut with a Leica Ultracut microtome. Prior to ultrastructure analysis in a Tecnai Sphera G² (FEI company, Eindhoven, Netherlands) at 120 kV, sections were contrasted in a LEICA EM STAIN with uranyl acetate for 30 min followed by an incubation in Reynolds' lead citrate for 1.5 min.

**Flow cytometric analysis and flow sorting of nuclei**. To estimate the number of Bs in individual Ae. speltoides plants, flow cytometric genome size measurements were performed. For this, 0.5 cm² of young leaf tissue was chopped with a sharp razorblade with equivalent amounts of leaf tissue of Secale cereale, subsp. cereale (R 737 from IPK-Gatersleben Genebank) as reference standard using the CyStain PI Absolute P reagent kit (Sysmex-Partec, Germany) according to the manufacturer's instructions. The resulting nuclei suspension was filtered through a 50 µm CellTrics filter (Sysmex-Partec, Germany) and the absorption distribution was measured on a CyFlow Space flow cytometer (Sysmex-Partec, Germany). The populations of nuclei were identified and gated in a dotplot fluorescence intensity versus side scatter (Supplementary Fig. 10) and the genome size and therewith the number of Bs calculated based on the values of the G1 peak means in the corresponding histogram. To analyse the proportion of B-chromosome-containing cells in 17–20 days old developing embryos nuclei were isolated and measured as described above. Slides with nuclei of root and leaf tissue of wheat-Ae. speltoides hybrids were prepared according to Hesse et al.[43]. In brief, plant tissue was fixed in 4% formaldehyde in Tris buffer (10 mM Tris, 10 mM Na2EDTA, 100 mM NaCl, 0.1% Triton X-100; pH 7.5) for 20 min on ice in a vacuum centrifuge. After washing twice in ice-cold Tris buffer, the tissue was chopped in nuclei isolation buffer (15 mM Tris, 2 mM Na2EDTA, 0.5 mM spermine tetrahydrochloride, 80 mM KCl, 20 mM NaCl, 15 mM β-mercaptoethanol, 0.1% Triton X-100; pH 7.5). The suspension was filtered through a 50 µm CellTrics filter (Sysmex-Partec, Germany), stained with 1.5 µg/ml DAPI and 2C nuclei were flow-sorted into Eppendorf tubes using a BD Influx cell sorter (BD Biosciences, USA). Equal amounts (12–15 µl) of sucrose solution (40% sucrose in Tris buffer; pH 7.5) and sorted nuclei suspension were pipetted on glass slides, gently mixed, air-dried overnight and kept at −20 °C.

**Whole-genome shotgun sequencing**. Library preparation (Illumina TruSeq DNA Sample Prep Kit) using genomic DNA isolated from leaves of 0B and 2B (referred as +B) plants from the Katzir population and sequencing (paired-end, 2 × 101 cycles, Illumina HiSeq2500 device) involved standard protocols from the manufacturer (Illumina Inc., USA). In total, 88.2 Gb of raw sequence reads were produced (Supplementary Table 1). Raw sequence reads were subjected to a trimming and filtering process using clc_quality_trim which cuts off bases with insufficient quality (Q-value <20) and subsequently removes short reads (min. length 50 bp), reads with adapter contamination and reads duplicated during PCR library construction. In total, 92.2% of reads and 91.0% of residues passed the post-processing (Supplementary Table 9). Validity was finally crosschecked by FASTQC[44].

**Sequencing of microdissected DNA and data pre-processing**. An Illumina NGS library was prepared from 23 pooled microdissected Bs using the PicoPLEX DNA-seq kit (Rubicon Genomics, USA). After the addition of 8% PhiX DNA, the library was sequenced (paired-end, 2 × 100 cycles) using the Illumina HiSeq2500 device according to the manufacturer's instructions, resulting in 2.4 Gb paired-end sequences (Supplementary Table 1). Following the recommendations from Rubicon Genomics (http://rubicongenomics.com/products/picoplex-dna-seq-kit/) we applied FASTQC tool[44] to identify biased sequences (read pairs). These were excluded together with the first 15 bp of each sequence read using cutadapt[45]. Raw data from the microdissected Bs were further processed in 4 steps using a workflow we refer to as the MiDiSeq pipeline (Supplementary Fig. 11). This workflow contains the following steps: first, adapter, low-quality and erroneous sequence fragments were removed using Trimmomatic[46]; second, screening for sequence contaminations was applied using BLASTN with default settings to identify and remove reads with critical hits to non-target sequences such as non-plant molecules or the PhiX reference (PhiX spiked-ins were used for sequence validation control). A reference alignment was performed as the third step to further enrich for the

target species. Here, BWA-MEM[47] was used to align pre-filtered reads from microdissected Bs to the de novo-constructed WGS assembly of *Ae. speltoides*. Captured reads were utilized by SAMtools[48] and in the fourth step extracted quality improved reads (1.55 Gb) into a FASTQ file by seqtk (https://github.com/lh3/seqtk). Finally, we reran FASTQC to screen for the validity of reads and found no skewed reads. To estimate the non-redundant length of the chromosomal segment captured by microdissection we assembled the cleaned FASTQ by CLC assembly cell (5.0.5) using default parameters. In total, 1.98 Mb of sequences were assembled consisting of 11,794 sequence contigs with an average size of 168 bp.

**De novo genome assembly**. Filtered and quality-enriched 0B and +B WGS sequences were merged and assembled using CLC assembly cell version 5.0.5. The de novo assembly pipeline was applied with automatic detection of the optimal parameters by CLC assembly cell, except parameters –b set to 60 and read distances in the –p parameter set to fb ss 100 450. All contigs below a length threshold of 200 bp were removed. Contaminations were identified by a BLAST + analysis[49] using NCBI nt[50] discarding non-plant-specific contigs. A total of 14,783 contigs were removed from the WGS assembly.

**Gene prediction and functional annotation**. In order to detect B chromosome-derived genic sequences we first applied gene prediction methods within our de novo constructed WGS assembly of the entire *Ae. speltoides* genome. For gene prediction, the GeMoMa[51] tool was applied using the reference genomes and gene annotation files (GFF) of five different species (*Brachypodium distachyon, Sorghum bicolor, Oryza sativa, Aegilops tauschii,* and *Triticum aestivum*). Subsequently, the species-specific predictions were merged and filtered for quality using the GAF filter. Gene prediction resulted in 29,228 putative gene models, of which 32% had homologs in at least two species. These gene models were distributed among 25,526 WGS contigs.

The predicted gene models were used to perform a functional annotation using two independent methods: InterPro[14] and a BLASTP analysis using 42,913 peptides of the wheat B subgenome including their corresponding functions available from the TGAC reference[15]. InterPro was applied with default parameter settings. The approach identified matches for 75.9% (22,173) proteins. We selected the best matches based on assigned e-values and required that a match exceeds at least 30% of the protein length. The final InterPro dataset assigned functions to 54.7% of our gene models. In the BLASTP approach, the E-value threshold was increased to E-10, other settings were default. Moreover, we retained only those hits showing greater than 50% of identity to the respective wheat protein and at least 50% of the query length. This approach enabled to assign functions to 74.8% (21,855 proteins) of our putative gene loci. Our final set relies on a comprehensive approach that combined both, the BLASTP and the InterPro methods. In total, 82.1% of the predicted *Ae. speltoides* gene models were assigned with a putative function. Final GO terms were checked with agriGO 2.0[52] to identify a correlated pathway.

**Assignment of WGS contigs to the wheat B-subgenome**. Assignment of assembled contigs to the B subgenome of *Triticum aestivum*[53] was done using BLASTN (-word_size 42; -evalue 1e−20). Results with % identity > =90 were further filtered selecting the best hit (highest bit score) that was accepted if a minimal alignment length of 500 bp was reached or the alignment length exceeded 90% of the query size. In the B chromosome composition analysis, we relaxed the post-processing of the previously-established BLASTN results. Multiple hits were allowed during this process due to the repetitive nature of the *Ae. speltoides* genome. A dynamic cut-off was used integrating multiple hits which fulfilled criteria to exceed 80% of the best detected bit score.

**Assignment to organelle genomes**. To assess the contribution of the cytoplasmic organellar genomes to the evolution of the B chromosome we included the complete chloroplast (NCBI KJ614406.1) and mitochondrial genomes (AP013107.1) of *Ae. speltoides*. All contigs with hits larger than 1 kb were assigned to the respective organellar compartment by BLASTN (default parameter).

**Identification of shared and B chromosome-specific sequences**. Two independent methods were applied to capture sequence contigs from the constructed WGS dataset that are shared by A and B chromosomes. First, we adapted the coverage ratio analysis approach[16] using *k*-mer frequency ratios and second, B chromosomal contigs were filtered by alignment of the microdissected B sequences (MiDiSeq). The first approach was performed with *Kmasker plants*[17] using the compare function that provided additional characteristics used to screen for sequence contigs with diverging *k*-mer patterns (with $k = 21$). Three of these characteristics were evaluated as criteria in the identification of B chromosome sequences. First, we determined whether the average *k*-mer frequency is slightly increased (termed KI33 for *k*-mer increase by 33%). Second, we determined whether the average *k*-mer frequency of a sequence is increased by at least 2-fold (termed AFC2 for average 2-fold-change). Third, we questioned whether any sequence contained sub-segments that are increased by at least 5-fold (termed FC5 for 5-fold-change). Thus, together with information gathered from the MiDiSeq evaluation, B chromosome assignment relied on four criteria that were defined to

express the likelihood that a contig belongs to the B chromosome. For all criteria a weighting was defined which expresses the reliability that a sequence can be assigned to the B chromosome (Supplementary Table 3). The final confidence is defined by the sum of individual criteria observed for a single sequence contig. Confidence classes 1–4 (1: low; 2: conservative; 3: moderate, and 4: high) are considered as evidence of derivation from the B chromosomes (Supplementary Table 3). The second approach (MiDiSeq) used the quality-enriched micro-dissected dataset. Due to the prevalence of repeats in the *Ae. speltoides* genome, we ran the read alignment using BOWTIE2 with very sensitive settings and filtered for uniquely mapped reads. In total, 27,143 WGS contigs were captured by more than 5 reads.

To investigate sequence similarity of B chromosome-derived sequences we aligned the merged (0B and +B) WGS assembly to the WGS assembly from the 0B dataset by BLASTN. For a stringent overlap calculation between the datasets we included only BLAST hits with the best bitscore. Subsequently, reads from microdissected DNA were aligned against the 0B WGS assembly using BOWTIE2 and the alignment was used for SNP calling applying samtools/bcftools[54]. We filtered only for SNPs in the derived overlapping regions using BEDTools[55].

To identify B-specific sequences, we screened all contigs assigned to Bs with a confidence class of 3 or 4. We applied the comparative *k*-mer approach of *Kmasker plants* using 21-mer indices of quality processed WGS of +B and 0B data. Subsequently, we looked into positions that were absent from the 0B *k*-mer sequence space. For an accurate estimation, we required that at least 80% of the contig length was absent in 0B, while the same contig was required to be present for more than 80% of its length in +B data. Contigs characterized by this criterion were defined as B-specific.

**Identification of potential B chromosome genic sequences**. We used sequence information from microdissected DNA together with a less stringent *k*-mer ratio analysis to detect gene fragments on the B chromosome. On the gene level, we specifically checked whether the respective diverging *k*-mer pattern (with $k = 21$) was located within the potential coding sequences to reduce assignment bias of B-derived genes. An increase of the *k*-mer frequency by 2-fold was used as a threshold to predict that a gene was present on the B chromosome. Reliability was further increased because enriched microdissected reads were required to align within the coding as well as the non-coding gene space spanned by introns. If both criteria were fulfilled, the gene was approved as B-derived. A total of 229 genes passed all criteria listed above and built our final set of candidate genic sequences assigned to the B chromosome of *Ae. speltoides* (Supplementary Data 1).

**A- and B chromosomes similarity**. As a basis for sequence similarity comparisons, we first identified regions where microdissected DNA from Bs overlap with the de novo assembly of the 0B dataset. These we termed B-related homologs. To achieve this, reads were aligned by BWA[47] and regions with sufficient depth of coverage (5-fold) were extracted using BEDTools[55]. Subsequently, these were linked to our final genome assembly of *Ae. speltoides* (0BAB) to reveal the overlap B-assigned contigs as well as B candidate genes. This was done by BLASTN with default parameter settings taking only the best hit into account. Homologous triplet regions were then analyzed for sequence variations using bcftools (1.9)[54] with two independent analyses using WGS reads from 0B dataset first and then micro-dissected sequences from Bs (Supplementary Table 7).

**GO term enrichment analysis**. The list of candidate genes (229) assigned to the B chromosome of *Ae. speltoides* (Supplementary Data 1) was subjected to gene ontology (GO) enrichment analysis using a singular enrichment analysis (SEA) tool of the agriGO v2.0[52]. The GO terms describing biological processes, molecular functions, and cellular components and associated with the candidate genes were retrieved from the InterPro and NCBI non-redundant databases. The list of B chromosome-specific genes was compared with all annotated genes (from 29,228 putative gene models) based on the two annotation strategies described above. A typical cut-off value of FDR < 0.05 (Benjamini FDR correction) was used in the multiple comparison correction process.

**Identification of root-related genes based on the BARLEX**. The list of candidate genes (229) assigned to the B chromosome of *Ae. speltoides* was subjected to the Barley Draft Genome Explorer (BARLEX, Colmsee et al.[56]). B-chromosome-related genes of *Ae. speltoides* (229) were aligned against all transcript sequences of high-confidence genes of barley (39,734 sequences; May 2016). Barley genes expressed specifically in roots were used to reveal potential root-specific genes within the 229 genes identified in the B chromosome (Supplementary Data 2 and 3).

**qRT-PCR analysis**. Differential expression of four root-related candidate genes of *Ae. speltoides* identified based on the BARLEX database was validated using qRT-PCR. RNA was extracted from roots of plants without B and with 4 B chromosomes. Additionally, leaves from plants without B and with 2 B, 4 B and 6 B chromosomes were collected. Material of five plants was pooled to form one biological replicate. Total RNA extraction was performed using the Spectrum plant total RNA Kit (Sigma-Aldrich Chemie GmbH, Germany) according to the manufacturer's manual. The RNA preparations were controlled for quality using a

NanoDrop One spectrophotometer (Thermo Fisher Scientific, USA). Reverse transcription was performed using a first-strand cDNA synthesis kit with oligo dT (18-mer) primer (Fermentas, Thermo Fisher Scientific, USA) and 2 μg of total RNA as a starting material. Amplification and detection of amplicons by qRT-PCR was performed on the QuantStudio 6 Flex system using the POWER SYBR Green Master Mix (Applied Biosystems, Thermo Fisher Scientific, USA). The cycling conditions comprised 10 min polymerase activation at 95 °C and 40 cycles at 95 °C for 3 s and 60 °C for 30 s. Two biological replicates per condition, each controlled with three technical replicates, were screened during the same run. The *GAPDH* reference gene[57] was used as an internal control to normalize the data. Relative gene expression was calculated using the comparative $2^{-\Delta\Delta CT}$ method[58]. Information about all primers used in the qRT-PCR analysis can be found in Supplementary Table 10.

**Reporting summary**. Further information on research design is available in the Nature Research Reporting Summary linked to this article.

## Data availability

Data supporting the findings of this work are available within the paper and its Supplementary Information files. A reporting summary for this article is available as a Supplementary Information file. The datasets generated and analyzed during the current study are available from the corresponding author upon request. Genome data were deposited in the European Nucleotide Archive (ENA) under project numbers PRJEB29864 and PRJEB29862. For transparency and reproducibility of the research process, main data objects are maintained as the following DOIs in the PGP repository[59]. The DOIs were build using the e!DAL system[60]. *Ae. speltoides* WGS assembly can be downloaded at https://doi.org/10.5447/ipk/2020/9, including the predicted gene models and functional annotations. Full information on the WGS contigs assignments and classifications are available at https://doi.org/10.5447/ipk/2020/11. Details of detected repeat copy numbers are available at https://doi.org/10.5447/ipk/2020/10. The functional annotation of 229 candidate genes from the *Ae. speltoides* B chromosomes and the GO term enrichment analysis are available at https://doi.org/10.5447/ipk/2020/8. The source data underlying Figs. 2e and 4, as well as Supplementary Figs. 3d, 5, 6a and 9 are provided as a Source data file.

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

## Acknowledgements

We thank Anne Fiebig, Anja Hartmann, Chris Ulpinnis, Katrin Kumke, Sylvia Swetik, Irem Aycan Sirel and Ines Walde (IPK) for technical assistance, Julie-Sophie Himpe (IPK) for art work, Armin Meister for support with statistical data analysis, as well as Ingo Schubert and Jeremy Timmis for insightful discussions. This work was supported by the Deutsche Forschungsgemeinschaft DFG (Grant No. HO1779/26-1, HO1779/30-1, SCHO1420/2-1) (to A.H., U.S.) and the China Scholarship Council (Grant No. 201606910015) (to D.W.).

## Author contributions

A.R. and D.W. performed embryogenesis and chromosome analysis and crossing experiments; A.Him. performed sequencing; T.S., A.B., and U.S. conducted bioinformatic analysis; V.S. performed super-resolution microscopy and centromere measurements; M.M. performed electron microscopy; J.F. conducted flow cytometry; M.R. and A.R. performed tissue culture experiments; A.B. performed qRT-PCR experiments; A.R., T.S., J.F., K.P., A.B., U.S., and A.H. analyzed the data; A.H., A.R., T.S., and U.S. wrote the paper.

## Competing interests

The authors declare no competing interests.
