## [Peer Review File · Nature Communications]

Reviewers' comments:

Reviewer #1 (Remarks to the Author):

This paper details the phenomenon of B chromosome loss in the root tissues of *Aegilops speltoides*. The data are quite clear for these claims. A clever experiment of inducing root differentiation and finding the loss of the B's under this circumstance illustrated a differentiation of roots being a trigger. The cytology and chromosome analyses are excellent.

A low pass sequencing effort led to the identification of a few hundred genes being present on the B chromosome. These were validated by comparing to sequences from slide isolated B chromosomes subjected to sequencing. Paralogues were sought in the B genome of wheat (related to *A. speltoides*). The claim is made that the B chromosome is derived from portions of all host chromosomes based on these widespread similarities. It is hard to conceive how this might happen and, indeed, the authors did not try. This is probably for the best—just report the homologies and let it rest at that.

The authors suggest that root elimination might be due to expression of genes in this organ that are detrimental. This is reasonable but there are only a very small fraction of genes predicted to be expressed in roots and there are no data indicating that these genes actually are expressed there. Moreover, there might be other scenarios such as selection for B maintenance in shoot tissues. This comment is a triviality but the authors might make their suggestion as "a reasonable suggestions among other possibilities".

The paper is very nicely written.

Reviewer #2 (Remarks to the Author):

In the manuscript entitled "The supernumerary B chromosomes of *Aegilops speltoides* undergo precise elimination in roots early in embryo development ", the authors provided detailed cytological evidence to show in goatgrass that the elimination of B chromosomes is a controlled and efficient root-specific process. At the onset of embryo differentiation, B chromosomes undergo elimination in proto-root cells. B chromosomes formed micronuclei, whose DNA was degraded eventually leading to B chromosome elimination. The authors used sequencing data with or without B content to identify the composition of B DNAs, and further found high confidence B-DNAs and B-genes. The methods used in the manuscript look valid. However, based on the proposal in the Abstract that "process might allow root tissues to survive the detrimental expression, or overexpression of 23 B chromosome-located root-specific genes with paralogs located on standard A chromosomes", I suggest the authors should provide more details about the functions of these 23 genes, or at least a specific gene example, based on sequence analysis and functional annotations, which were not described in the Results, to further discuss potentially what detrimental effects these genes might have.

Reviewer #3 (Remarks to the Author):

The manuscript on B chromosome elimination from the roots of a wild diploid wheat is from the lab of an expert on both B chromosomes and chromosome elimination. It presents a series of careful observations to show that the elimination of Bs is strictly controlled and highly efficient, and that it is a root-specific process. Many lines of evidence come together, carried out with high quality experiments: for example, the authors compare embryogenesis with adventitious root formation - the latter showing less efficient B-chromosome elimination; and they look at CENH3 centromere signals to show centromeres are not inactivated. Assays are then able to show how DNA is fragmented in B-chromosome micronuclei as the final step leading to loss of the B chromosomes. A robust

bioinformatics approach is used to identify B chromosome specific genes and sequences, notably better than various approaches over the last decade because of use of more sequencing data, exploitation of high-quality wheat sequence assemblies, and better tools, although still not leading to a very high quality B-chromosome assembly. The analysis showed the genes present on B chromosomes, and that 10% showed root-specific expression patterns.

The extensive discussion makes the most of the data. Careful comparison shows many similarities with the meiotic drive and non-disjunction, but not including spindle asymmetry, that the authors have shown for B chromosomes at meiosis. Nevertheless, more cell biology and protein analysis/localization will be needed to complement the work presented here to show detailed mechanisms.

The abstract is not entirely suitable and in some cases unnecessarily over-sells the work. For example, "Here, we present the first analysis of programmed chromosome elimination in plants." is not entirely true given the authors and others extensive work on gametocidal chromosomes, genome elimination in hybrids, previous B chromosome drive work, and some apomictic meiotic-replacement/embryosac situations.

I am glad that authors speculate on why B chromosomes are eliminated from the roots: "This process might allow root tissues to survive the detrimental expression, or overexpression of 23 B chromosome-located root-specific genes with paralogs located on standard A chromosomes." However, as far as I can see, this explanation would only hold if the corollary, of advantageous expression of B chromosome genes in the shoot, were to occur: otherwise, the elimination mechanism need not be root-specific.

Response to referees:

Reviewer #1 (Remarks to the Author):

This paper details the phenomenon of B chromosome loss in the root tissues of *Aegilops speltoides*. The data are quite clear for these claims. A clever experiment of inducing root differentiation and finding the loss of the B's under this circumstance illustrated a differentiation of roots being a trigger. The cytology and chromosome analyses are excellent.

A low pass sequencing effort led to the identification of a few hundred genes being present on the B chromosome. These were validated by comparing to sequences from slide isolated B chromosomes subjected to sequencing. Paralogues were sought in the B genome of wheat (related to *A. speltoides*). The claim is made that the B chromosome is derived from portions of all host chromosomes based on these widespread similarities. It is hard to conceive how this might happen and, indeed, the authors did not try. This is probably for the best—just report the homologies and let it rest at that.

ANSWER:

Based on the observed similarity between A and B chromosome-located coding sequences we propose that the B chromosome of *Ae. speltoides* represents a 'by-product' of the host genome evolution. Similar conclusions about the likely origin of Bs were drawn for the Bs of vertebrates and other phylogenetic groups (eg. Banaei-Moghaddam et al., 2015; Makunin et al., 2016; Valente et al., 2014). However as noted by the reviewer, we did not elaborate on this claim as the available genome sequence information is not sufficient for an in-depth analysis of the mechanism behind the *de-novo* formation of a B chromosome.

Reviewer #1 COMMENTS

The authors suggest that root elimination might be due to expression of genes in this organ that are detrimental. This is reasonable but there are only a very small fraction of genes predicted to be expressed in roots and there are no data indicating that these genes actually are expressed there. Moreover, there might be other scenarios such as selection for B maintenance in shoot tissues. This comment is a triviality but the authors

might make their suggestion as “a reasonable suggestions among other possibilities”. The paper is very nicely written.

ANSWER:

We agree, we can only speculate on why the B of *Ae. speltoides* undergoes strict elimination in roots. It would be informative to manipulate the expression of candidate root-specific genes but unfortunately, no protocol is available for the transformation of *Ae. speltoides*.

The suggested ‘selection for B maintenance in shoot tissues’ is another option which we have addressed in the DISCUSSION part of the manuscript. Now it reads: ‘We suggest that root-specific elimination of Bs is required to remove B chromosome-located genes which, if expressed in roots, would lower the inheritance of the selfish chromosome because of reduced plant vigor. The possibility that positive selection for B chromosome maintenance in shoot tissues has occurred remains to be investigated.’

To strengthen our observation about the identification of possibly root-specific genes we performed qRT-PCR with a subset of paralogous genes encoded by A chromosomes. Due to the limited quality of the available genomic sequence of *Ae. speltoides*, the absence of sufficient sequence polymorphisms between gene family members and the presence of numerous multi-member gene families (see Supplementary Data 1), specific PCR primers could be generated for only 4 of the 20 candidate genes. All the genes analysed revealed preferential expression in roots. The absence of Bs in roots prevented further testing the potential B-located root-specific candidate genes.

Reviewer #2 (Remarks to the Author):

In the manuscript entitled "The supernumerary B chromosomes of *Aegilops speltoides* undergo precise elimination in roots early in embryo development", the authors provided detailed cytological evidence to show in goatgrass that the elimination of B chromosomes is a controlled and efficient root-specific process. At the onset of embryo differentiation, B chromosomes undergo elimination in proto-root cells. B chromosomes formed micronuclei, whose DNA was degraded eventually leading to B chromosome elimination. The authors used sequencing data with or without B content to identify the composition of B DNAs, and further found high confidence B-DNAs and B-genes. The methods used in the manuscript look valid. However, based on the proposal in the

Abstract that "process might allow root tissues to survive the detrimental expression, or overexpression of 23 B chromosome-located root-specific genes with paralogs located on standard A chromosomes", I suggest the authors should provide more details about the functions of these 23 genes, or at least a specific gene example, based on sequence analysis and functional annotations, which were not described in the Results, to further discuss potentially what detrimental effects these genes might have.

ANSWER:

We applied more stringent BLAST search conditions resulting in a reduction in the number of root-specific candidate genes from 23 to 20 (see provided Supplementary Data 1). Unfortunately, we are not aware of a root phenotype caused by misregulation of any of the identified candidate genes. Because of this, we prefer to refrain from predicting possible detrimental effects on root development of any individual gene.

To strengthen our observation about the identification of possibly root-specific genes we performed qRT-PCR with a subset of paralogous genes located on A chromosomes. Due to the limited quality of the available genomic sequence of *Ae. speltoides*, the absence of sufficient sequence polymorphisms between gene family members and the numerous multi-member gene families (see Supplementary Data 1) specific PCR primers could be generated for only 4 of the 20 candidate genes. All these genes showed preferential expression in roots. The absence of Bs in roots precluded testing the identified B-encoded root-specific candidate genes.

Reviewer #3 (Remarks to the Author):

The manuscript on B chromosome elimination from the roots of a wild diploid wheat is from the lab of an expert on both B chromosomes and chromosome elimination. It presents a series of careful observations to show that the elimination of Bs is strictly controlled and highly efficient, and that it is a root-specific process. Many lines of evidence come together, carried out with high quality experiments: for example, the authors compare embryogenesis with adventitious root formation - the latter showing less efficient B-chromosome elimination; and they look at CENH3 centromere signals to show centromeres are not inactivated. Assays are then able to show how DNA is fragmented in B-chromosome micronuclei as the final step leading to loss of the B chromosomes. A robust bioinformatics approach is used to identify B chromosome specific genes and sequences, notably better than various approaches over the last

decade because of use of more sequencing data, exploitation of high-quality wheat sequence assemblies, and better tools, although still not leading to a very high quality B-chromosome assembly. The analysis showed the genes present on B chromosomes, and that 10% showed root-specific expression patterns.

The extensive discussion makes the most of the data. Careful comparison shows many similarities with the meiotic drive and non-disjunction, but not including spindle asymmetry, that the authors have shown for B chromosomes at meiosis. Nevertheless, more cell biology and protein analysis/localization will be needed to complement the work presented here to show detailed mechanisms.

The abstract is not entirely suitable and in some cases unnecessarily over-sells the work. For example, "Here, we present the first analysis of programmed chromosome elimination in plants." is not entirely true given the authors and others extensive work on gametocidal chromosomes, genome elimination in hybrids, previous B chromosome drive work, and some apomictic meiotic-replacement/embryosac situations.

ANSWER:

We did not intend to over-sell our work and therefore revised the sentence. Now it reads: 'Here, we present a detailed analysis of programmed B chromosome elimination in plants'.

Reviewer #3 (Remarks to the Author):

I am glad that authors speculate on why B chromosomes are eliminated from the roots: "This process might allow root tissues to survive the detrimental expression, or overexpression of 23 B chromosome-located root-specific genes with paralogs located on standard A chromosomes." However, as far as I can see, this explanation would only hold if the corollary, of advantageous expression of B chromosome genes in the shoot, were to occur: otherwise, the elimination mechanism need not be root-specific.

ANSWER:

We have toned-down and extended our statement about the likely reason behind the root-specific elimination process of Bs. Now it reads: 'We suggest that root-specific elimination of Bs is required to remove B chromosome-located genes which, if expressed in roots, would lower the inheritance of the selfish chromosome because of

reduced plant vigor. The possibility that positive selection for B chromosome maintenance in shoot tissues has occurred remains to be investigated.'

References

- Banaei-Moghaddam, A.M., Martis, M.M., Macas, J., Gundlach, H., Himmelbach, A., Altschmied, L., Mayer, K.F., and Houben, A. (2015). Genes on B chromosomes: Old questions revisited with new tools. *Biochim Biophys Acta* 1849, 64-70.
- Makunin, A.I., Kichigin, I.G., Larkin, D.M., O'Brien, P.C., Ferguson-Smith, M.A., Yang, F., Proskuryakova, A.A., Vorobieva, N.V., Chernyaeva, E.N., O'Brien, S.J., *et al.* (2016). Contrasting origin of B chromosomes in two cervids (Siberian roe deer and grey brocket deer) unravelled by chromosome-specific DNA sequencing. *BMC Genomics* 17, 618.
- Valente, G.T., Conte, M.A., Fantinatti, B.E., Cabral-de-Mello, D.C., Carvalho, R.F., Vicari, M.R., Kocher, T.D., and Martins, C. (2014). Origin and evolution of B chromosomes in the Cichlid fish *Astatotilapia latifasciata* based on integrated genomic analyses. *Mol Biol Evol* 31, 2061-2072.

REVIEWERS' COMMENTS:

Reviewer #1 (Remarks to the Author):

I am satisfied with the revision.

Reviewer #2 (Remarks to the Author):

I have no further questions.

Reviewer #3 (Remarks to the Author):

The authors have made accurate and helpful changes to the manuscript. The lab is undoubtedly the world's expert group on supernumerary B chromosomes, and in this work, they address a long-standing question about a diploid wild wheat grass, *Aegilops speltoides*, where these B chromosomes are eliminated in the root but maintained in the shoot. Careful cell biology is used to show the programmed and now well-defined processes of elimination in the newly differentiating root meristem in the embryo. While the B chromosomes are not essential, the work provides an interesting comparator for the other (important but specialized) systems where chromosomes show programmed elimination, or even abnormalities/diseases where chromosomes are gained or eliminated. As well as the very significant cell biology, the work also contains a valuable analysis of chromatin and DNA sequences in the eliminating micronuclei containing B chromosomes and the nature of the genes present on the B chromosomes, speculating which genes may be deleterious in roots but not shoots. The corollary, that B chromosomes are advantageous in the shoots (in other species, the whole organism), has been discussed many times previously.

The work uses appropriate methods, including exploiting the materials developed in the authors' lab such as probes and antibodies. The new root-specific expression of paralogous genes to those from the *Aegilops* B chromosome in barley *Hordeum* is clear and certainly makes the conclusion regarding interference with the root development processes much more robust.

The final model is convincing (Fig 4) and placed well in the context of insect literature with B chromosome elimination. The paper is very concise, and while I might have chosen additional or a different illustration from the supplementary material, perhaps reducing the Methods parts which are largely standard, the main paper is clear while those with particular interest will certainly go to the Supplementary. The video is attractive but largely reinforces the data from still micrographs. There will be interesting future work on both DNA methylation and chromatin protein modification, alternative methods to silence genes.

The paper is extremely well written and I have no specific comments.

REVIEWERS' COMMENTS:

Reviewer #1 (Remarks to the Author):

I am satisfied with the revision.

Reviewer #2 (Remarks to the Author):

I have no further questions.

Reviewer #3 (Remarks to the Author):

The authors have made accurate and helpful changes to the manuscript. The lab is undoubtedly the world's expert group on supernumerary B chromosomes, and in this work, they address a long-standing question about a diploid wild wheat grass, *Aegilops speltoides*, where these B chromosomes are eliminated in the root but maintained in the shoot. Careful cell biology is used to show the programmed and now well-defined processes of elimination in the newly differentiating root meristem in the embryo. While the B chromosomes are not essential, the work provides an interesting comparator for the other (important but specialized) systems where chromosomes show programmed elimination, or even abnormalities/diseases where chromosomes are gained or eliminated. As well as the very significant cell biology, the work also contains a valuable analysis of chromatin and DNA sequences in the eliminating micronuclei containing B chromosomes and the nature of the genes present on the B chromosomes, speculating which genes may be deleterious in roots but not shoots. The corollary, that B chromosomes are advantageous in the shoots (in other species, the whole organism), has been discussed many times previously.

The work uses appropriate methods, including exploiting the materials developed in the authors' lab such as probes and antibodies. The new root-specific expression of paralogous genes to those from the *Aegilops* B chromosome in barley *Hordeum* is clear and certainly makes the conclusion regarding interference with the root development processes much more robust.

The final model is convincing (Fig 4) and placed well in the context of insect literature with B chromosome elimination. The paper is very concise, and while I might have chosen additional or a different illustration from the supplementary material, perhaps reducing the Methods parts which are largely standard, the main paper is clear while those with particular interest will certainly go to the Supplementary. The video is attractive but largely reinforces the data from still micrographs. There will be interesting future work on both DNA methylation and chromatin protein modification, alternative methods to silence genes.

The paper is extremely well written and I have no specific comments.

AUTHOR RESPONSE:

Many thanks for the kind and supportive comments.